# The Sacralization of Politics? A Case Study of Hungary and Poland

**Joanna Kulska**

Institute of Political Science and Administration, University of Opole, 45-061 Opole, Poland; jkulska@uni.opole.pl

**Abstract:** Religion influencing politics and politics impacting religion to achieve its own, very non-religious, goals are determining the reality of contemporary states and of global politics. Mutual relations between religion and nationalism have proven to be one of the most complex and unequivocal challenges shaping contemporary states in the areas of both their domestic and their foreign policies. This article is an attempt to compare two cases which are often wrongly perceived as twin models of links between religion and politics, namely Poland and Hungary. In both states, based either on actually present or on "constructed" Christian values, myths and symbols religious–national narratives have been developed by leading politicians (Jarosław Kaczyński in Poland and Viktor Orbán in Hungary) linked directly to the sacralization of ethnos (nation) and the ethnicization of religion. The conducted analysis has a theoretical character. The sacralization of nation and the ethnicization of religion occurring in Poland and Hungary are presented against quite different historical and cultural backgrounds and levels of religiosity/secularization in both countries. In order to explain this specificity, an analysis is performed upon a broader historical and cultural context and upon a specific understanding of religion and nationalism in Central and Eastern Europe.

**Keywords:** sacralization; (chosen) nation; nationalism; political strategy; secularization

## 1. Introductory Remarks

In spite of having a very different background, including historical, cultural and political traditions as well as various models of relations between religion and politics that developed over the centuries, similar phenomena can be observed in many parts of the world. Religion is increasingly back, and religion is increasingly politicized. As such it is becoming an agent of change (Berger 1997) or is at least engaged in exerting some political influence respective or irrespective of political theologies (Toft et al. 2011) and taking the form of, among other things, religious nationalisms. As a matter of fact, this characteristic of religious resurgence has become one of the most omnipresent demonstrations of the significance of the religious factor in the political sphere and is also the key determinant questioning the accuracy of secularization theory. As Christopher Soper and Joel Fetzer point out in their seminal study, the politics of nationalism has been very much affected by the renaissance of religion that has taken place over the last few decades, contrary to the predictions of secularization theory (Soper and Fetzer 2018, p. 7).

Religion influencing politics and politics impacting religion to achieve its own, very non-religious, goals are determining the reality of contemporary states and of global politics. Mutual relations between religion and nationalism are proving to be one of the most complex and unequivocal challenges shaping contemporary states in the areas of both their domestic and their foreign policies. When they are analyzed, the question of new forms of sacralization or resacralization of politics is raised. Yet, in many cases, these forms in fact represent the return to well-known schemes which turn out to be very lively and very effective in spite of the long and profound secularization processes. As such, they concern not only countries where the Christian heritage has dominated the religious landscape but also those grounded in other major world religions (Berger 1999; Haynes 2020; Barrow 2021;

Cesari 2022). Religion that "refused to stay at home" (Casanova 1994) shows its power of influence where it historically or socially has been the crucial determinant of a political culture, a concept of the nation or an everyday axiology, such as in Poland or in secular states of the West such as the US or France. Additionally, however, it has turned out to be a useful and effective political instrument in constructing some new, and not rarely quite effective, religious–political narratives, as is happening in Hungary.

From the scholarly point of view, these developments are an additional argument in favor of the critics of the Western, dichotomic view on relations between religion and politics based on the assumption that the "either/or" scheme is the only one possible, and in which no overlapping between the spheres should be taken into consideration (Wilson 2012; Habermas 2008; Hurd 2007). They also contribute to the debate on the validity and accuracy of the secularization theory as such (Habermas 2008; Taylor 2007; Martin 2005; Karpov 2010; Fokas 2009; Burchardt et al. 2015; Obirek 2021). This debate, though not new, is now increasingly enriched with the question of the links between nationalism, religion and identity, pointing heavily to the recognition of identity as one of the main drivers of global political developments (Fukuyama 2018). In this discussion, attention has been paid to the limited but also limiting application of the secularization theory both in the theoretical and empirical dimension. Central and Eastern Europe (CEE) with its own, non-Western concept of the nation, nationalism and identity seems to be one of the "beneficiaries" of this redefined approach, opening new levels of analysis that overcome these limitations and allow for a more comprehensive understanding of the religious factor in global politics.

This article is an attempt to compare two cases which are often wrongly perceived as twin models of links between religion and politics or more precisely the politicization of religion, namely Poland and Hungary. In both states, based either on actually present or on "constructed" Christian values, myths and symbols, religious–national narratives were developed by leading politicians (Jarosław Kaczyński in Poland and Viktor Orbán in Hungary) linked directly to the sacralization of ethnos (nation) and the ethnicization of religion (Zenderowski 2010). The main aim of this strategy has been the introduction of a nationally oriented, religiously based ideological alternative against a "Western model of decay" (Riccardi 2022, pp. 88–102) serving mainly internal political goals.

The conducted analysis has a theoretical character. The sacralization of nation and the ethnicization of religion occurring in Poland and Hungary are presented against quite different historical and cultural backgrounds and levels of religiosity/secularization in both countries. In order to explain this specificity, an analysis is performed upon the broader historical and cultural context and specific understanding of religion and nationalism in Central and Eastern Europe. The article stipulates that top-down sacralization is serving as a "salvation of nation" in the context of the sacred legitimization and missionary role of the "chosen nation" which is the background strongly present in the region of Central and Eastern Europe and which differentiates this part of the continent from its Western part. At the same time, Catholic nationalism that is "utilized" (Poland) or "invented" (Hungary) is applied as a political tool in the conditions of bottom-up secularization. As a result, selective processes of sacralization and secularization are simultaneously present, leading to emerging politicized and pseudo-religious—in this case pseudo-Christian—patterns. This results in further secularization of those segments of a society that are critical towards the politicization of religion. Furthermore, it results in the deepening of normative conflicts within societies and the weakening of constructive religious potential (Campbell 2020).

## 2. Theoretical Background

A scholarly debate on mutual links between nationalism and religion has been developing over many years. Three major approaches have emerged in this debate over the last few decades. According to modernist theories, nationalisms have been historically constructed and have been products of modernization with religion playing little or no part in forming national consciousness. Secular nationalism replaced ethnic and religious

identity for national identity based on civic and secular norms and triumphed over privatized and depoliticized religion. According to ethno-symbolic theories, nationhood was based on preexisting religious ties that political elites rediscovered and applied in the forms referring to ethnic heritage and traditions. Religion has played a vital role in the origins of nationalism, and continues to do so, being an important element of national identity. These two approaches thus assumed an essential difference and potential for competition between civic and spiritual identity. However, scholars have also offered a third model that refers to Durkheim's view, and according to which nationalism is itself seen as a secularized form of religion (Soper and Fetzer 2018, pp. 3–8).

Durkheim's notion that the understanding of the sacred can change but will never disappear situates itself in the center of the considerations on religious nationalism. It also provides a foundation for the type of nationalism that is neither religious nor secular, or is both, combing modernist and ethno-symbolic models. According to Durkheim, every society reaffirms its unity by referring to something sacred, understood as superior and absolute, and the general dichotomy between the sacred and the profane is always reestablished. He argued that while secularization was historically inevitable, something still had to play the integrative role performed by religion in the pre-secular society. Given the necessity of meaning and purpose as well as myths and symbols associated with it, in the secularized societies nationalism would be this component (Buchenau 2015, p. 261; Soper and Fetzer 2018, p. 8).[1] According to José Casanova, who focuses on this phenomenon from the contemporary perspective "( . . . ), obviously some secular reality (the nation, citizenship, the individual, inalienable rights to life and freedom) can be sacred in the modern secular age" (Casanova 2012, p. 453). At the same time, however, "( . . . ) the modern "secular" is by no means synonymous with the "profane" nor is the "religious" synonymous with the modern "sacred"" (Ibid., p. 460).[2]

Anthony Smith observes that if we adopt a more functional and Durkheimian perspective, we may perceive in nationalism a particular form of 'political religion' whose tensions with traditional religions have led to two important consequences: a growing politicization of religion and the messianisation of politics and the elevation of the 'people'. These are both the result of strains that occur when nationalisms invoke or ally themselves with particular religious traditions. Based on that, a number of transformations characteristic of the modern world are produced which contribute to the sense of flux and mass instability. According to this author, the politicization of religion means that traditional motifs are endowed with new political significance (Smith 2000, pp. 792–99). This phenomenon may occur both in the moderate and extreme forms and refers to the situation when traditional religions serve legitimate social order, a particular regime or political community against destructive forces (Zúquete 2017, p. 446).

Messianisation of politics, whereby the nation and its leaders are exalted and endowed with religious charisma, is the reverse process to the politicization of religion (Smith 2000, p. 799). With the messianisation of politics, some special, extraordinary mission is appointed to the nation, legitimizing its status as the "chosen one". A related term that can be used to analyze how nationalisms and religions are connected is also the sacralization of politics. Emilio Gentile observes that sacralization of politics (variously defined as lay religion, secular religion, earthly religion, political religion, political mysticism, and political idolatry) takes place when, more or less elaborately and dogmatically, "( . . . ) a political movement confers a sacred status on an earthly entity (the nation, the country, the state, humanity, society, race, proletariat, history, liberty, or revolution) and renders it an absolute principle of collective existence, considers it the main source of values for individual and mass behavior, and exalts it as the supreme ethical precept of public life. It thus becomes an object for veneration and dedication, even to the point of self-sacrifice" (Gentile 2000, pp. 18–19). According to this author, the sacralization of politics, which has been both democratic and totalitarian and has taking the shape of democratic 'civil religion' and totalitarian 'political religion', means the formation of a religious dimension in politics that is distinct from traditional religious institutions. It occurs when politics

is conceived, lived and represented through myths, rituals and symbols which perform crucial functional roles. This is because people demand faith in the sacralized secular entity, dedication among the community of believers, enthusiasm for action, and a warlike spirit and sacrifice in order to secure its defense and its victory (Gentile 2000, pp. 18–55).

The relationship between political religion, sacralization of politics and politicization of religion is not an obvious one, and it seems those terms are used sometimes interchangeably. The specificity of political religion is in its giving of a sacred character to significant social entities, such as race or nation, but also the class or the human kind, which therefore become the object of reverence and devotion. The members of such an entity are considered the chosen community, and their political activities are perceived as fulfilling a messianic mission. It is important to stress that the sacralization of politics and political religion are not to be directly identified with the politicization of traditional religions, which is aimed at using religion for political purposes (Burgoński 2014, p. 222). Yet it is in fact not difficult to overtake and apply some elements of sacralization of religion in a political struggle which needs to be based on the creation of some meaning, purpose, myths and symbols. Sacralization is thus a concept which may or may not include an officially religious component, but in fact it is hardly possible to exclude it. National identity and religious identity in some cases can overlap or match to a degree that makes it impossible to differ between one from another. Religion can enhance some objective and subjective features of national identity, such as an idea of homeland, common myths, history, culture, communal rights and duties, customs or language. Belonging to a given ethnic or national group usually means a closer or more distant relation to the religious denomination and some religious institution that serves both as an integrative factor to the given community and a legitimizing factor to the political elites. Even in the case where highly secularized and ethnically diversified societies are based on the concept of secular nationalism, it is difficult to avoid some references to religious language, symbols or rituals (Zenderowski 2014, p. 305).

According to Anthony Smith, the relationship between religion and nationalism should be perceived at three levels. The first would be an official level supported by the state as well as cultural and political elites expressed in the form of official national symbols. The second one would be at a more grassroots, or folk, level and expressed in the form of the religious practices of a given society which affect how national identity is understood. This difference is important from the perspective of enhancing differences in the understanding of the concept of national identity and its links to religion between the "enlightened political elite" and the folk. The third level is what might be termed the sacred properties of the nation, or more accurately, the sources and properties of the nation conceived as a sacred communion of its members. In many ways, this is the most relevant and important for an understanding of the relationship between 'religion' and 'nationalism' within and between states, even in the secular West (Smith 2000, pp. 800–2).

Two interrelated processes may occur when religion and nationalism are combined, namely the ethnicization of religion and the sacralization of ethnos (Zenderowski 2010). Ethnicization of religion means the process where a given group usurps religious elements such as the figures of the saints, the places of the religious cult, the selected elements of the religious doctrine or the religious doctrine as such. This way religion, or rather its modified form, becomes "indivisible property" of the given nation. As religion is nationalized and loses its universalistic character, religious symbols become equally important to national symbols. In that case, religion gains a specific ethnic value and in fact becomes a national ideology, an element sustaining endangered identity (Posern-Zieliński 2005, p. 25). On the other hand, sacralization of ethnos could be understood as attributing to itself some religious meaning by the given ethnic group through which it gains the guarantee of immortality. Thanks to this process, it becomes a quasi-religious community that gathers past, present and future generations. This given group, the nation, which is perceived as the chosen nation, is predestined to fulfill a special mission, usually a difficult one requiring an extraordinary sacrifice that other groups do not need to fulfill. This way, the

nation is sacralized and placed "higher" compared with the other, "not chosen" groups that do not have a special mission to be fulfilled (Smith 2003, pp. 19–43; Zenderowski 2014, pp. 306–7).

Both the ethnicization of religion and the sacralization of the nation may have consequences for religion itself. This problem is especially challenging from the perspective of Christianity, which is not only based on the idea of universalism but also proposes some clear pattern of relations between religion and politics. In Christianity, the politicization of religion should not take place, just as the existence of political religion should not, as such, take place. Political religion is based on the vision of an "earthly" realization of the perfect order. As in Christianity, this perfect order can be fully achieved only in eternity because Christianity "desacralized" politics through the introduction of the dualistic relations between religion and politics. Christianity separated what is eschatological and what is political (Burgoński 2014; Casanova 2012; Gray 2021). In reality, however, religious doctrine and political application of this doctrine may differ significantly (Toft et al. 2011). The concept of nationalism which, as already mentioned, is in the practical sense inseparable from religious connotations, may become both the subject of self-instrumentalization and instrumentalization (Zenderowski 2014, p. 304). In this last case, it is the area of politics in which this instrumentalization often takes place, incurring serious harm for the religion itself and the religiosity of the given society and also resulting in the processes of faster secularization of those segments of the society that are critical towards this kind of entanglement between religion and politics.

The sacralization of politics, occurring in the form of the ethnicization of religion and the sacralization of the nation has become one of the key factors determining and legitimizing national discourses in Poland and Hungary. Over the last two decades, in both countries, one encounters an increasingly advanced model of selective "overtaking" of religion by politics but not in the sense of normative frames outlined by religious axiology and ethics. This phenomenon instead represents a political strategy of emphasizing some of the most "fitting" elements of religious doctrine to strengthen political arguments while abandoning or even openly opposing the others.

When looking at the sacralization of politics, a crucial issue seems to be the question of who is actually usurping religion, how, and in which direction this usurpation is developing. Taking a broader look allows for the proposal that this usurpation can be differently legitimized. It can, namely, refer to the process of the bottom-up sacralization of the nation that has developed more "naturally", as an inherent element of the cultural tissue of a given entity, now "used" or instrumentalized by political elites as an element of political strategy, which would be the case of Poland. Yet it can also be the process of top-down sacralization which does include some marginal elements of historical and cultural tradition but is in fact "re-invented" or even created as key factor of political strategy, which would be the case of Hungary.

## 3. Poland and Hungary as "Two Brothers?"

The history of relations between religion and politics in Europe is obviously a very complex one. The love–hate relationship between religion and states has continued up until today and is present both through cultural embeddedness and through particular bonds with national identity (Fokas 2009, p. 402). As Michael Minkenberg observes in Central and Eastern Europe (CEE), conditions for the development of the reasserted role of religion in the political context are different here than they are in the West. In this part of the continent "( . . . ) secularization as a politically imposed process in the communist era does not need processes of pluralization or a growing presence of Islam to ignite religious fervor: the extreme anticommunist impulse of the radical right leads directly to the historical connection between religion and nation-building in the region" (Minkenberg 2018, p. 387; Pew Research Center 2017, 2018a). While the question of how the radical right should be defined and whether this concept refers to Fidesz and PiS is debatable, there is no doubt that, in this part of Europe, history matters (Hroch 2020; Grabowska 2018; Borowik and

Tomka 2001), especially when religious and national loyalties are discussed. Poland is one of such countries. Christopher Soper and Joel Fetzer refer to Poland in the post-communist era as an example of a country where religious and national loyalties have created one, coexisting phenomenon. In Hungary the situation is different. Here, a re-emergence of religious nationalism has been observed that is based on the surrogate of religion and is used as the skeleton for a newly constructed axiology (Soper and Fetzer 2018, p. 1; Ádám and Bozóki 2016). This differentiation between Poland and Hungary is of key importance for understanding the complexity of the CEE itself, which is often inaccurately perceived as a homogenous or almost homogenous one.[3]

The specificity of the region, in terms of the mutual relations between religion and nationalism, is the result of a "stormy history" that did not have much in common with the West. Looking into the most recent historical developments, the patterns developed during the times of communism need to be examined. In spite of the atheist ideology imposed across the whole Soviet bloc, the situation differed in various countries from the effective process of atheization to the preservation of religion as the fundamental reference point for the collective identity and political loyalty. However, one needs to realize that this relation goes much further than just the communist era, usually reaching the period of the Reformation but also the origins of those states which are usually deeply attached to the founding myth and first rulers under whom they were Christianized (Ramet and Borowik 2017; Grzymała-Busse 2015; Stan and Turcescu 2011).

The general characteristic of the region has been linked to the essential role of religious identification accompanying ethnic or national identification since the Middle Ages. "Contrary to the majority of states and nations of Western Europe on the lands situated between German and Italian states (from the West) and lands ruled by tsars/Russian imperators, the Mongol/Tatar chans and Turkish sultans religion has become the primary modern source of group identity ( . . . ) which with time resulted in different kinds of mergers with emerging national identities that stayed in stronger or weaker connection with the premodern ethnic identities" (Zenderowski and Pieńkowski 2020, p. 7; Pew Research Center 2017). The process of the emergence of the concept of the nation is incomparable with the one that took place in the West also due to other factors. When modern nation states arose in Western Europe as the top-down process of unification promoted by cultural and political elites, in Central and Eastern Europe the age of premodern multinational, multireligious and multicultural empires was the dominating reality. Moreover, in this part of the continent, nations created states and not the contrary as was the case of Western Europe. As a result, nations that inhabit this part of Europe can imagine their existence independently from the territories of the currently owned states and border lines (Zenderowski 2018, pp. 137–38).

Countries of the region can be grouped into three basic types representing different dependencies between religion and national identity. One type is formed by those nations which can be almost identified with one denomination, which is the case of Poles but also Croats, Serbs or Greeks. The second type is formed of multireligious nations, for which national identity arose as the more significant integrative factor than religion. Finally, the third type comprises those nations for which religion is not the relevant factor in their social life. For those nations—the Czechs, Estonians, Latvians and partially Hungarians—religion could be either removed from the public life or treated as a "decorative element" (Zenderowski and Pieńkowski 2020, pp. 7–10).

In the case of the first type of states, where one religion dominates, the ethnicization of religion and the sacralization of ethnos are present as the natural, bottom-up processes that developed along with the emergence of the given nation, with Poland being one of the examples. Here, one can speak of a specific kind of religious identification, one which is ascribed to individuals, rather than chosen by them, and experienced as fixed rather than as changeable (Jakelić 2006, p. 136). The understanding of mutual relations between nationalism and patriotism in CEE is also peculiar. As Timothy Garton Ash remarks, nationalism does not automatically mean negative, violent ethnocentrism but in fact patriotism (Garton Ash 2022, pp. 27–29), which is difficult to understand for representatives of the

West. Miroslav Hroch emphasizes the contextual view that is needed to comprehend this connection. Nationalism is not just the social fact, or objective belonging to the given community but the commitment that those societies recognize as an important reference point. In CEE, nationalism is not a private matter and not just a sociological phenomenon, but an internal duty (Hroch 2020) usually linked to the religious dimension.

It has been the societal power of religion, expressed in the correlation between patriotism and nationalism, that has defined the patterns of mutual relations between religion and political structures in CEE. Importantly, the loyalty emerging or not emerging as an element of national thinking was conditioned by the type of connection to political power in the previous decades and centuries. Conflicts between the nations and their secular opponents gave the Churches an opportunity to act as defenders of nations and national identity and cohesion. Where Churches shielded the nations, patriotism became inseparable from religious loyalty. In the course of those fierce struggles, national and religious identities melted together, forging a powerful form of religious nationalism (Grzymała-Busse 2015, p. 8).

Poland and Hungary are on opposing sides when this factor is taken into consideration. For Hungarians who underwent a shift towards Calvinism during the Reformation period, Catholicism was linked to oppressive Habsburg rule and not linked to true patriotism (Halmai 2018). Thus, in spite of the fact that Catholicism had been the religion of the majority of Hungarians right after WWII, it was not connected to national loyalty. One might risk the statement for this reason that it was easier for the communist government to subordinate religious institutions, not only Catholic but also those representing other denominations in Hungary (Enyedi 2003). During the communist era, in spite of the heroic attitude of cardinal Mindszenty (Schanda 2019), a fast and effective process of secularization developed.

This was not the case with Poland. Here, a true symbiosis between the nation and monotheistic religion was much more deeply rooted as the basis of folk religiosity.[4] As such, it became the key point of social integration and communal identity from the Partition time against the oppression exerted by stronger neighbors, especially Protestant Prussia and Orthodox Russia. Based on that, the concept of patriotism was closely connected with Catholicism (Gowin 1999; Kłoczowski 1978). As a matter of fact, and, again, contrary to a Hungary in which, during the Reformation period, Calvinism had gained some believers, in Poland the Reformation did not become deeply rooted and was quickly lost with the counter-reformation (Grabowska 2018, pp. 21–22; Stan and Turcescu 2011, p. 118)[5]. Additionally, it was the tradition of independence from the state structures that turned out to be of crucial importance. The alliance between the throne and the altar was not possible in Poland, as the throne was a foreign one (Taylor 2002, pp. 61–62). As a result, while in the West drastic secularization took place and in some CEE states effective atheization took place, Polish Catholicism went through an extraordinary revival at the end of communist period (Casanova 2004). In Poland the "more natural" nation–religion link was preserved, while in Hungary a deep process of secularization in the form of an imposed atheist ideology took over. Hence, during the communist era in Poland, the link between religion and the national identity was strengthened, while in Hungary it was weakened.

Although Poland and Hungary represent distinct cases in terms of the historical and cultural embedding of Catholicism and its links to nationalism and collective identity, some common features are present that have been similarly utilized for the ethnicization of religion and the sacralization of both nations. The concept of "the chosen nation" has been applied in both cases to carry out the mission of fulfilling the civilizational duty of the salvation of the "mistaken/wrong others" against the destructive forces of Western liberal values. Defining or re-defining such narratives would not be possible without referring to some specific historical developments that are now used to shape and re-shape historical memories. In both cases, a retrospective mentality, rather than prospective one, is becoming the essential base of a national consciousness that is typical of more traditional, past and history-oriented societies (Grondona 2003).

Their extended state traditions contrast Poland and Hungary from the rest of the region. Both Poland and Hungary were regional powers in the past. In regard to Poland, the reference is made to the golden age of Polish statehood in the XVI century and the Jagiellonian dynasty when the Polish–Lithuanian Commonwealth belonged to the biggest territories in Europe reaching almost 1 million km$^2$. In the case of Hungary, the tradition of the Hungarian Kingdom, reaching back to the tradition of Saint Stephan and lasting over 900 years from 1000 till 1918 is emphasized. Meanwhile, long periods of the non-existence or limited existence of statehood have marked the region, including Poland and Hungary. In the case of Poland, it was "only" 123 years, while in the case of Hungary it was 341 years. Thus, the concept of "liquid statehood" emerged as the reason for the linking of collective identity to the nation, more than to the state. The pattern of states constituted as a consequence of the "will of the nations" resulted in the "borderless" concept of a nation not coherent with the existing state borders, which is the case for both the Poles and the Hungarians. This also overlaps with the lack of respect for institutions, including that of the state. What counts is the community living by the strength of spirituality and not the strength of administrative effectiveness (Hroch 2020; Zenderowski 2018, pp. 138–39).

Additionally, Poles and Hungarians belong to the most homogenic societies in Europe (Laczo 2021). Importantly, homogenic nations of Poles and Hungarians which developed in the course of the XX century were not the result of internal political strategies but the change of borders decided by the external political players. In the case of Hungary, it was the consequence of World War I and the Treaty of Trianon. In the case Poland, it was the outcome of World War II, and more precisely the Yalta and Potsdam treaties. Deep victimization over the territories lost as a result of the decision of external forces, including those very historically important pieces of land, such as Transylvania for the Hungarians[6] and Kresy for the Poles, was the constitutive element shaping the national ideologies and mythologies and referred also to huge national minorities left outside of the borders. However, the list of sorrows is actually much longer. As Radosław Zendrowski remarks, thinking in terms of the categories of victim and tormentor, guilt and satisfaction are inseparable elements of public debates on state and nation in Central Europe. Being an inherent factor of retrospective mentality typical for the region, these categories have become elements of the ominous repertoire of accusations and demands in international politics. Moreover "( . . . ) we are witnessing a somehow bizarre victimization contest for which of the nations suffered the most" (Zenderowski 2018, p. 141). A specific feature of collective thinking is also the trauma of physical extermination, rapid depopulation due to mass emigration or "melting" into the closer or more distant and stronger ethnos (Ibid., p. 142). Interestingly, as Attila Pató notices in case of Hungary, "The conceptual elements of the national self-representation are often linked to the images of the survival of an ever-suffering nation in heroic victories and falls among great powers and adjoining nations, more often than not watched as potential adversaries—with Poland as the only real exception in the neighbourhood" (Pató 2018, p. 206).

All of these elements are strongly present in the "sacralized" discourse. As such, they refer to what is a relatively typical feature that characterizes the CEE region namely the mixture of pride and shame towards the external world. One of the dimensions of this approach is the concept of moral superiority towards the external offenders, which is the main element of the "vision of salvation" strongly present both in Poland and in Hungary (Zenderowski 2018). Messianism is an inherent element of this vision, directed both into the internal audience to gain political support of those most linked to national identification and to the outside audience to "save Europe", though the latter is rather in service to the former.

## 4. National Catholicism: Between Sacralization and Secularization

In spite of diverse historical developments, religious traditions and levels of religiosity, national Catholicism has become one of the main determinants of political discourse in Poland and Hungary over the last two decades (Riccardi 2022, pp. 88–102). In Poland,

described not only as "the Christ of nations" but also as "antemurale christianitatis"[7] (Davies 2022, p. 8), this may be understandable as a consequence of historical developments. Meanwhile, in Hungary, one can speak of the "re-invention" of national Catholicism in the form of the official state ideology performed by Viktor Orban for purely political purposes. This ideology, described as paganized Christianity (Ádám and Bozóki 2016), ethno-nationalism (Halmai 2018) or religious neo-nationalism (Povedák 2020), does not in fact have much in common with Christian values but is utilized as a key integrating and delimiting determinant (Héjj 2018). As Rene Nieland remarks: "( . . . ) Viktor Orbán uses a tool, which, at least in political discourse, had relatively little relevance in Hungary, yet now serves marvelously for his purpose of redefining Hungary and its international status: the reference to Christianity and Christian heritage of Hungarian history" (Nieland 2019).

This strategy, which can be perceived as one of many illiberal innovations (Enyedi 2020), is proving to be very effective. Orbán has been successful in creating a new identity based on religious affiliations in the non-religious society. In Hungary, where national Catholicism is neither the result of a bottom-up process of religious socialization nor the result of the initiative of the Church itself (Riccardi 2022, p. 92), it has become the major political reference point in the process of the sacralization of the nation. Such instrumentalization of religion and the shift towards political, top-down Catholicism embodying national values does have some historical roots. It served as an identity element in the interwar period during the Horthy regime (Mink 2019; Fazekas 2015) when the territory of Hungary was decreased by two thirds compared with the prewar period. This was the moment of the emergence of Hungarian messianism. The country was at that time compared to Christ and Trianon to cross. The idea of a reconstruction of the 1000-year-old monarchy of Saint Stephan become the absolute imperative of Hungarian thinking and was even expressed during everyday prayers at schools in the calls for a "resurrection" of "old Hungary" (Stachowiak 2020). This was also the strategic trial of the linking the nation to a Catholic religion which was not a historically national religion, in spite of its being the religion of the majority of Hungarian society. Moreover, Hungarian nationalism that has been developing for the last 150 years did not include a religious element as the crucial determinant (Pató 2018, p. 206). The bloody conflicts of the Reformation meant that, until the Horthy era, no Church could fully identify itself with the Hungarian nation. While more numerous and more powerful in comparison with Protestantism, the Catholic Church still played a small historical role in preserving national consciousness, and it never became equated with Hungarian patriotism. During communism, the Catholic Church neither served as a symbol of national independence nor as a source of protection for the opposition as had happened in Poland (Halmai 2018). As a result, during communism the Hungarian national memory was neither clearly Catholic nor Calvinist (Buchenau 2015, pp. 261–84). It did become "officially" Catholic in the period of the last two decades when Catholic rituals were found to be the proper rituals for strengthening collectivity (Enyedi 2003, p. 172) and reaffirming the sacred status of the nation. Despite being a Calvinist himself, Orbán has referred to Pope John Paul II as the "Holy Father" and stated once that the Hungarian government voted for Catholic values. He has participated in Catholic masses and processions and has proposed declaring a Catholic holiday, namely the Assumption of Mary celebrated on 15 August, as the national holiday—something which surprised even Hungarian Catholics (Enyedi 2003, pp. 172–75). As Zsolt Eneydi points out "( . . . ) most proclerical initiatives did not come from the churches but from the prime minister himself. ( . . . ) it was he, and not the churches, who proposed that the state should supplement the salaries and pensions of the priests" (Ibid., p. 172).

Viktor Orbán was able to create a new national Catholic ideological skeleton in a society which had never recovered from the communist secularization but that also did not show much attachment to religion in general, something which is clearly visible according to the sociological data. As Bulcsu Bognár and Zoltán Kmetty show in their study, the value orientations of religious people in Hungary appear to be aligned with secular society. According to their research, the main trend seems to be that the moral judgements of

religious people do not differ from those of the people who consider themselves as not religious or atheist. Religious people in Hungary no longer find guidance in their religious tradition—which is regarded as outdated in today's secularized public discourse—but are oriented by the values of the secularized world, drifting away from their Christian roots (Bognár and Kmetty 2020). In fact, in Hungary, declarations of belonging to Christian civilization usually stay in opposition to religious dogmas and ethics. Under the Danube, "Christianity" is a lockpick word in political campaigns, especially anti-immigrant ones, and in disputes with the European institutions (Héjj 2018). Yet, irrespective of the low level of religiosity and religious practices, it was "made" a foundation of sacralization of the nation. Additionally, another crucial element of collective identity, which remains in doctrinal conflict with Christianity but serves integrating purposes well, has been applied. This is namely the pagan element of Hungarian identity that is also willingly used by Viktor Orbán in the form of the "Turul bird",[8] the symbol of pre-Christian, ancient Hungarians (Ádám and Bozóki 2016, p. 109). With the simultaneous application of this pre-Christian heritage, the instrumentalization of Christianity seems to be even more evident.

In Poland, where Catholicism was strongly preserved "in times of trial", meaning the Partitions, WWII and the communist era (Grabowska 2018, p. 23), an "entangled", or "twisted" national–religious identity survived as one of the main features characterizing Polish society (Koseła 2003; Łuczewski 2012). Yet starting from the first years of the transformation period some clashing visions on the role of the Catholic Church emerged. For the Church, confrontation with the phenomena and processes described as modernity (Gowin 1999, p. 26) turned out to become a challenge; one that resulted in increasing criticism, especially within left-liberal political circles. The voices questioning the authority and special social and political role of the Church were more often heard but also the notion of "the Pole–the Catholic" started to evolve towards quite a diversified phenomenon.[9] In spite of that, the general public's view on the essential role of Catholicism for self-identification dominated until at least 2010, meaning the Smoleńsk crash (Wigura and Terlikowski 2022, p. 204).[10]

Interestingly, the inclusion of Christianity, and more precisely Catholicism, as a political resource, took place in a similar period and in a similar way. After the transformation period in the 1990s, both Viktor Orbán and Jarosław Kaczyński were far from referring to religious arguments when developing their political strategies. At that time, Fidesz used to be a liberal party with militantly anti-clerical views (Halmai 2018; Enyedi 2003). The shift was quite sudden. For Orbán, it can be dated from the mid-90s, and for Kaczyński from the beginning of the 2000s when he openly admitted the change of ideology and the turn towards a more conservative, closed wing of Polish Catholicism embodied by Father Rydzyk and Radio Maryja[11] (Pankowski 2010, p. 153).[12] For both of them, Catholicism became, at the time, a reservoir of conservative values which were to become the foundation for the creation of nation-oriented, anti-European, and antiliberal political stances. For Hungary, the culmination of this process was the adoption of the new Constitution in 2011 where this new role of the CC has been defined as preserving the nationhood based on the millennial tradition of the Crown of Saint Stephan. In Poland, this has been rather a process marked with some breakthrough points, including the death of John Paul II in 2005, the crash of the presidential plane in Smoleńsk in 2010 and controversies over the so-called presidential cross used as a political symbol, and finally the direct alliance between the throne and the altar (in spite of the opposition of some bishops) after the elections of 2015 (Sowiński 2014). According to those visions, Poland can survive only as a Catholic country. The famous words that Jarosław Kaczyński addressed during the 24th birthday of Radio Maryja, only a few weeks after the victory in the parliamentary elections in 2015, need to be perceived as the motto of his political program: "The hand raised against the Church is the hand raised against Poland!" (Kaczyński 2015). The remarkable continuation of this view was the one expressed by Jarosław Kaczyński in 2019 that outside of Church there is only nihilism which builds nothing and destroys everything (Kaczyński 2019).

When comparing the data concerning Hungary and Poland, visible discrepancies are present. According to the Atlas of European Values in 2017, 45% of Hungarians said religion was very or quite important to them, while in the case of Poland it was almost 79%. As for religious services in Hungary, only 10.30% of respondents declared attending religious services at least once a week, while in Poland it was 47.70% (Atlas of European Values 2017).[13] However, this has been changing quite rapidly. For decades Poland was considered a special case, being the example of a country where secularization processes had been taking place exceptionally slowly in reaction to progressing modernization (Casanova 2004; Ramet and Borowik 2017; Grabowska 2018). Recently, which means specifically in the period of the last 5–7 years, secularization has been accelerating, and in some age groups even exceptionally. Strategic, political sacralization is thus accompanied by bottom-up secularization.

The most visible change in the Polish religious landscape is the radical secularization of the youngest generation; one that had been "crawling" for some years but has recently reached the highest level in the global scale according to Pew Research Center findings (Pew Research Center 2018b). The rate of those between 18 and 24 years old regularly attending religious services dropped from 69% to 23% in the period of the last 30 years (Grabowska 2022). In 2021, religious education was attended only by 52% of students at secondary school level and, according to Mirosława Grabowska "( . . . ), both the inheritance of religious attitudes between parents and children and the conscious formation of faith in the family are weakening. What we observe at the parish level and in families makes us think that secularization processes will not slow down" (Ibid.). In fact, it was the moment of an open reaching out to the religious–national discourse after the elections of 2015 that overlapped with the acceleration of the process of secularization that was made visible by the drops in religious practices, the number of seminarists and in trust towards the church (Riccardi 2022). On the other hand, the process of secularization is developing slower in the older generations of Poles. As Mirosława Grabowska has remarked, older generations are affected by two "vaccines": the memory of communism and the direct influence of John Paul II, which makes these generations more "immune" to secularizing factors. Meanwhile for the youngest generation of adults (18–24) "it is already the history, if not archeology" (Grabowska 2022).

According to the results of the all-Polish survey summarizing Synod consultations, the most challenging problems of Polish Catholicism are clericalism, the lack of communication between the clergy and the ordinary members of the Church, hermetic language on the side of the clergy, the low quality of homilies, and the absence of women and the young generation (EKAI 2022). Yet another important controversy, which is the issue of heated debate in Poland, is the problem of relations between the ruling government and the Polish Catholic Church; a relation which, for many Poles, is far too close. A large number of Catholics directly blame the Church for speeding up the secularization process (Zoll 2022; Dylus 2018, pp. 135–36). Opinions are held that consider Jarosław Kaczynski to be not talking about Jesus but to be talking often about the role of the Church as a political tool. Gladly accepting the political commitment of many bishops and the majority of Polish clergy to PiS, he seems to be saying: "No to God and yes to Church". Meanwhile those Catholics who negatively assess the political activity of priests and Church institutions seem to be saying "Yes to God, No to Church" (Makowski 2021).

## 5. Churches at the Service of God or Politics?

Despite their many differences, Poles and Hungarians are very close in opinion regarding formal relations between religion and politics: the church and state should be separated according to 70% of Poles and 67% of Hungarians (Pew Research Center 2018a). Obviously, how separation/autonomy/neutrality should be understood and what model of separation is the most expected is one of the most challenging problems of not only CEE but basically all European states (Ferrari and Cristofori 2016; Torfs 2010; Madeley and Enyedi 2003; Minnerath 2021). Poland and Hungary are not any exception.

Churches can become politicized in two ways: they can make a decision to become players in the political field or they can simply allow themselves to be infiltrated in their own sphere and become factionalized according to their political sympathies. At the same time churches are given privileges by the state in accordance with the legitimacy they can provide to the political elite (Grzymała-Busse 2015; Enyedi 2003, pp. 163–73). Mutual entanglement between Churches and the political power in Poland and Hungary do not represent the same model. The Polish Catholic Church being a huge and internally divided structure seems to be following all of the above scenarios, preserving its basic independence and political influence. In Hungary, Churches have been subordinated by political power following the path known from more distant history. While Poland's Catholic Church is an active player on the political scene, in Hungary, Churches are, in the majority of cases, passive supporters of government.

As Zsolt Enedi stresses, in Hungary the main Churches are very clearly associated with the governing party with some minor differences present. Traditionally, the nationalist attitudes were strongest in the Calvinist Church while the Catholic Church is typically much less nationalist in rhetoric than the Calvinist Church. However, when it comes to the statement of individual bishops, the large majority of bishops functioning now in Hungary have made very clear political statements. The Lutherans, who are a very small community, are different. They are basically the only ones who say "no" to this Christian–nationalist discourse. Sometimes loudly, sometimes in a more cautious way, but they have a different orientation (Matlak and Enyedi 2021). The Catholic Church in Hungary does not speak out on moral issues, nor is it involved in the migration crisis. It spoke clearly only once, in 2015 at the height of the crisis, when Cardinal Péter Erdő, the primate of Hungary, called for tougher government policies and for closing the borders to refugees, believing that by accepting immigrants, the Church itself would become a people smuggler (Héjj 2018).

At the same time, the Primate of Poland, Archbishop Wojciech Polak, wrote on one of the social networks: "We are called to recognize the face of Christ in refugees, because "I was a newcomer"" (Ibid.). In fact, the migration issue was not the only one to raise official critics of the Church towards PiS. What differs the Polish case from the Hungarian one are the moments of open opposition of the Polish Episcopate when the politicization goes too far. At least a few examples could be mentioned here. In 2006 the Episcopate criticized the stabilization pact between PiS and Radio Maryja, even though in 2005 their internal instruction encouraged members to vote for right wing parties. In 2010, it opposed the instrumentalization of the so-called "Smolensk cross" that was installed in front of the Presidential Palace shortly after the Smolensk plane crash in which president Lech Kaczyński with his wife Maria and representatives of Polish elite died. According to the official statement the most important Christian symbol was used by politicians as a means of political dispute (Sowiński 2014, p. 681). In 2017 a document entitled *Christian shape of Catholicism* was prepared under the auspices of the Polish Episcopate raising, among others, the necessity of differentiating between true patriotism and exclusive nationalism as well as the endangered Polish–German reconciliation, which were great achievements initiated by Polish Bishops in 1965 and were now being "reversed" (Konferencja Episkopatu Polski 2017).

## 6. Concluding Remarks

Poland and Hungary are examples of states where entanglements between nationalism and religion cannot be seen as objective, politically neutral processes of sacralization of both nations. Instead, these are strategically planned developments serving both internal and external political purposes occurring in the form of the simultaneous resacralization and secularization of both societies. In quite a brief period of about two decades, religion has been "politically used" as an element of political strategy in order not only to gain and sustain political power but also to conduct ideological revolution against the existing cultural and political order (Riccardi 2022).

At the same time, Poland and Hungary, though usually "put into the same basket" show significant differences when analyzed more profoundly. In the case of Poland, resacralization appeals to deeply rooted tradition, while in Hungary it entails the process of the newly designed/initiated patterns. As a "side effect" it also incurs the institutional entanglement of authority of the Churches. In Poland this results in the model of a balance between the mission and the interest of the Catholic Church (Kulska 2021) which has benefitted from the alliance between "the throne and the altar" and from the nation-oriented preferences of the dominant part of the Polish Catholic clergy (Makowski 2021). In the case of Hungary, this implies the subordination of the main Churches to the ruling political elites and making them objects that play the role, with some marginal exceptions, of the "humble servants" on the side of "Caesarian domain" (Sata and Karolewski 2020).

Ambivalent, complex interaction between nationalism and religion in Hungary and Poland appears to have many consequences. As shown in this paper, it turns out to be a very efficient political tool; one that proves the ever-present potential of religious–national links. It is the ability of religion not only to evoke "collective effervescence" but also to strengthen the "tribal thinking" which is willingly supported or even re-invented by political elites, not always for the benefit of society nor for the benefit of religious values. In this case, normative conflicts within societies between more liberal and more conservative segments of the society can be deepened and can be effectively used to gain more devout political followers. In Hungarian society, the religious–national rhetoric appears to be helpful in winning the support of such voters and dominating the political scene in spite of long-established secularization. In Poland, in spite of an extraordinary decrease in religiosity and a speeding-up of secularization processes, the religious–national narrative remains crucial for preserving the support of about one third of voters; those that are the most devout. Ethnicization of religion and the sacralization of the nation is exactly what those voters need. Through these processes the mixture of pride and shame characterizing the CEE region as such has been operationalized. The power of collective trauma has been strengthened by the power of collective messianism.

Yet these entanglements also have consequences for the religiosity of the society and religious doctrine itself. On the one hand, it contributes to the process of secularization of less nationalistically oriented segments of the society critically assessing religious institutions moving too far away from their spiritual goals including especially the youngest generations, which is the case of Poland divided into a more open, inclusive and more traditional, exclusive Catholicism. On the other hand it, is religion itself losing its "prophetic potential" (Campbell 2020)[14]. As Stanisław Obirek observes, using religion to build strong ethnic identity usually ends not only in the gradual washing out of religion from this identity, but also in the building of ideological meanings on it, often contradictory to the values of the professed religion (Obirek 2022).

When observing the political developments in Poland and Hungary over the last two decades, it is not possible to predict whether the links between "the religious" and "the national" will become any less important in the decades to come, even in the case of political reconfigurations. Contrary to that, religious–nationalistic orientations will be referred to as normative and integrative frames in the conditions of an increasing feeling of insecurity, a weakening of social collectiveness and a rise in identity anxiety. These phenomena, which are obviously not the reality of only the CEE countries, enhance the invention and re-invention of patterns of cultural, social and political self-identification, and are difficult to define without reference to the key dimensions of collective identity, including religion and the nation. This way, a more or less religious sacralization of the nation and ethnicization of religion will continue to be a part of the renaissance of identity and the raising level of emotions characterizing contemporary politics (Fukuyama 2018; Moïsi 2010). As such, following Durkheim's notion that the sacred can change but will never disappear, they will be present and politicized irrespective of progressing secularization.

**Funding:** This research received no external funding.

**Data Availability Statement:** Not applicable.

**Conflicts of Interest:** The author declares no conflict of interest.

## Notes

[1] Charles Taylor observes that today's secular world is characterized not by an absence of religion by rather by constant emergence of some new options, both religious and non-religious, which give meaning and normative frames to individuals and collectives (Taylor 2007).

[2] José Casanova distinguishes between three basic terms: secular, secularization and secularism. Secular has become a central modern category to construct and understand a reality differentiated from "the religious". Secularization refers to actual or alleged patterns of transformation and differentiation of the institutional spheres of "the religious" and "the secular" from early modern to contemporary societies. Secularism in turn means a whole range of modern secular worldviews and ideologies that may be consciously held and explicitly elaborated into normative–ideological state projects (Casanova 2009, pp. 1050–51).

[3] In the conventional wisdom there is the conviction of the close relations between both nations expressed in the slogan "The Pole, The Hungarian, two brothers, to the sable and to the glass".

[4] It is important to note that Poland also had a tradition of being a multiethnic and multireligious state that started to develop in the Middle Ages including the settlement of Jews who arrived to Poland already in the XI century. In 1573, the Warsaw Confederation (Compact of Warsaw), one of the first acts granting absolute religious freedom to all non-Catholics in Poland, was adopted by the Sejm. The historical change towards exposing the role of Catholicism took place in the XIX century, meaning the moment of losing statehood. This is when the pattern, or as some authors say, the stereotype of "The Pole–The Catholic" (Polak–Katolik) emerged. Two lines of thought are regarded as especially important in terms of combining the concept of Polishness and Catholicism, namely the national ideals developed during Romanticism (Poland as "the Christ of nations") and the national ideology of Roman Dmowski, one of two key political figures that shaped Polish independence in 1918. Roman Dmowski, who himself was not a religious person, considered Catholicism not as an addition to Polishness but as an element which constitutes its essence. He was the one who created the concept of "The Pole–The Catholic" as part of political discourse (Ciunajcis 2018, pp. 163–64; Obirek 2022; Wigura and Terlikowski 2022, p. 205).

[5] In the case of Poland, these were Jesuits who managed to effectively reverse the Reformation wave in the XVI century.

[6] Apart from Transylvania, some other historically important territories lost as a result of the Treaty of Trianon are now an important element of Viktor Orbán's politics (Upper Hungary, Carpathian Ruthenia) raising tensions between Hungary and its neigbours (Romania, Slovakia and Ukraine).

[7] Norman Davies observes that Poland as an "antemurale christianitatis" ("Bulwark of Christianity") was dominated for over 1000 years by Roman Catholicism and as such it was the barrier against the forces of Islam and Eastern Orthodoxy (Davies 2022, p. 8).

[8] The Turul bird is one of the most mystical symbols in Hungarian mythology reaching back to prehistoric Hungary. It is the relic of ancient faith, the embodiment of the superior powers and monarchical sovereignty. It is the symbol of national identity, togetherness and of the Holy Spirit that protected Hungarians in the ancient times (Béni 2016).

[9] Jan Szczepański observed back in the 1970s that the notion "The Pole–The Catholic" does not mean strictly following religious doctrine and ethics but expresses itself rather in the more ritual dimension (Szczepański 1971, p. 288). Based on this observation and the conducted research, Beata Pawłowska proposes five types of "The Pole–The Catholic": fanatic, believing, façade, declarative and ex. (Pawłowska 2015, pp. 89–90).

[10] Sociological data show that from 1989 till 2015, Poles were quite stable in their opinions on the relations between the Church and the state. They accepted the traditional model of the endorsed Church, meaning separation yet, at the same time, special recognition of Catholicism and the Catholic Church. They perceived the presence of religious symbols and rituals in the public sphere as something natural. They were quite critical on the moral teaching of the Church (such views were expressed by 33% of respondents in 2015). At the same time, an overwhelming majority rejected political engagement of the Church (Grabowska 2018, pp. 202–11). According to Karolina Wigura and Tomasz Terlikowski, the moment of the presidential plane crash in Smoleńsk in 2010 means the beginning of the slow separation between Polishness and Catholicism (Wigura and Terlikowski 2022, p. 204).

[11] Radio Maryja became the key creator of the national–Catholic ideology (Krzemiński 2017, pp. 85–112) in the post-transformation period and engaged in close relations with the parties expressing national–Catholic views. Established in 1991 by charismatic leader father Tadeusz Rydzyk, it grew into a conservative community known as the Radio Maryja Family. Mirosława Grabowska observes that this community could be considered an equivalent of the American religious right. It is strongly socially rooted and has significant mobilization capacity, including electoral participation. For this reason they constitute an influential social and political factor (Grabowska 2018, p. 214).

[12] In the same speech, Jarosław Kaczyński made some remarks of Viktor Orbán and presented himself as less nationalistic: "( . . . ) you have to look for a broader formula. You have to shoot with the guns that are available. ( . . . ) First, I have to win elections. For this reason I move to the right as much as I can, not as far as Orbán in Hungary, he took over an extreme nationalist electorate,

but still. You cannot win elections without Radio Maryja. Once I wanted to do it in another way. The Centre Alliance was an attempt to base oneself on the centrist voters. It ended up as a failure". See (Pankowski 2010, p. 153).

13      According to the Pew Research Center it is 41% for Poland and 9% for Hungary as far as weekly attendance in concerned. See (Pew Research Center 2017). https://www.pewresearch.org/religion/2017/05/10/religious-commitment-and-practices/ (accessed on 30 December 2022).

14      David Campbell refers to evolving role of mutal relations between religion and politcs in the US, yet the consequences of entanglement between religion and politics seem to have much in common when Christian countries are discussed.

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
