# Peer review of "The Sacralization of Politics? A Case Study of Hungary and Poland"

_religions, doi:10.3390/rel14040525_

Round 1

Reviewer 1 Report

The publication is of a very high standard, I congratulate you on your choice of topic and encourage further research.

Author Response

Dear Reviewer

Thank you very much for your positive, encouraging review.

Reviewer 2 Report

Formal page

The author uses the abbreviation CCE without clarifying it.

Abbreviation WWII more suitable WW II

Line 344: Hungry corrected to Hungar

Material page

Why did the author decide to write the article?

I don't think it's that much of a hot topic, a topic that would resonate with a lot of people.

However, it is a topic that definitely needs to be brought to the scientific and professional community.

Therefore, it can be easily stated that the article, or his subject, has great potential. This is why I didn't reject the article per se. I assure the author that I am intimately familiar with this issue and a more thorough revision of the topic is also my wish.

The author should not make an attempt to compare, but make a real in-depth analysis of the issue.

In the case of Hungary, is it possible to build what Prime Minister Viktor Orbán is doing on other than Catholic soil?

Viktor Orbán is doing on other than Catholic soil?

The attempt to present Calvinism as a religion closely connected with Hungarian identification was, rather, in the territories that were "torn away" from the so-called "Great Hungary“.

In my opinion, the author makes a too extensive introduction to the issue, I recommend shortening this part, especially the ideas that are repeated in some places.

Miklos Horthy also followed the St. Stephen's tradition, i.e. the Catholic one (even if the author mentions it briefly), an example is his speech in Kasssau (Košice). The union of church and nation in Poland is also a big topic. The author mentions e.g. Transylvania, he does not mention Felvidék, he does not mention other territories, he does not clarify the issue as a whole. I recommend the author to do a much deeper analysis.

The issue of the terms Magyarország and Hungary is not clarified. (Magyarország vs. Hungary, Ungarn, etc. Ugorshchina (Ukrainian), Uhorsko (Slovak), Uhersko (Czech)). How do Hungarians perceive him? Isn't Viktor Orbán using this topic?

In one sentence, the author mentions pre-Christian symbols, such as the bird Turul, but it does not even briefly clarify what it is connected with, what kind of mythical bird it is. An ordinary reader cannot know what it is about. (As it is related to, for example, Almos, I am not asking for an in-depth analysis, rather a brief explanation). It is the same in the case of Father Rydzik, Radio Mário and the closed wing of Catholicism, what is it about? It needs at least a brief explanation.

The author does not analyze why such actions are taken by leading politicians both in the case of Poland and in the case of Hungary. What are they looking for?

Finally, such a comparison should not miss what such actions of politicians can lead to. I recommend that the author add a special section to the article in which he presents his perspective on what it can lead to?

The President of the Russian Federation, Vladimir Putin, presented himself similarly and based his campaigns on similar slogans....

Reviewer 3 Report

1. In future steps' research it could be interesting to discuss your analysis considering both (the already considered) classical secularization theory (secularisation as substitution - Durkheim - or Aufhebung - Hegel -) and secularization as primacy of the functional differentiation of society (Luhmann). Cfr. Diotallevi, Ordine imperfetto, 2014.

2. In future steps' research it could be interesting to consider the two kinds of western "disciplinary revolutions" (Gorski) bottom-up and top-down.

3. It would be very interesting considering the Italian case as one made up of many and very differents pre-unitarian religion-politics models just one of these leading the (very delayed) state formation process. 

Author Response

Dear Reviewer

Thank you very much for your positive review and some valuable suggestions for the future.

Reviewer 4 Report

Thank you for submitting this paper for review. I was happy to see an article providing some nuance to the debates about Central-Eastern Europe and the differences between the contexts of particular countries, in this case, Hungary and Poland. However, while the argument concerning the difference between the two countries is interesting, the paper does not make it compelling for several significant reasons:

1.      Lack of conciseness - the paper lacks conciseness, as it introduces many “filler” sentences that do not add any value to the argument. Already the first sentence of the paper is generic “In spite of a very different background, including historical, cultural and political traditions as well as various models of relations between religion and politics that developed over the centuries, similar phenomena can be observed in many parts of the world.” Other examples: “A scholarly debate on mutual links between nationalism and religion has been developing over many years.” “The history of relations between religion and politics in Europe is obviously a very complex one.

2.      Lack of conceptual clarity – the paper uses different concepts in an unclear manner, most significantly “religion” and “religious,” which is in some way differentiated from “politics,” but the precise difference between the two is unclear. It comes with certain presumptions that are not made explicit in the paper. For example, the Catholic Church is described as “politicized,” which could mean that it was not politicized before or that it is not political on its own. Probably a more apt term, in this case, would be “instrumentalized,” which the authors use in other places.

3.      Essentialization of Polish nationalism – this is one of the most problematic aspects of the paper. While the constructed character of Hungarian nationalism is described at length by the authors, the complexity of Polish nationalism is flattened and presented as, at least to a certain extent, a given. The authors ignore the large part of the debate on nationalism concerning the development of Poland and Polish identity as a multi-religious and multi-ethnic entity, simply describing Polish identity as tied to one denomination. Instead of arguing that, as the authors put it, the connection between Catholicism and Polishness developed “naturally,” a more careful argument would point out a more long-standing tradition of equaling Polishness and Catholicism while bringing critical voices to the table as well.

4.      Undisciplined structure of the argument - at this stage, the paper is too undisciplined to argue carefully for the case it is making. It jumps from one topic to another without going deep enough into any, nor without properly engaging with the literature on topics that are extremely broad – from secularization and the “return of religion” (which should also be treated more critically) through nationalism to historical constructions of the “imagined identities” and the politics of memory. Because of that, instead of deconstructing the existing biases and providing more nuance, the paper reaffirms some of the existing biases (e.g., “Polak-Katolik”). Similarly, the conclusions of the paper refer to the notions that are not discussed in the main body of the paper (e.g., concerning the influence of the relationship between religion and nationalism on secularization).

Thus, the paper needs to be completely restructured and rewritten. The paper needs to make a smaller, more focused, and more disciplined argument that is constructed with more precision and greater detail, including the critical gaps mentioned above.

Round 2

Reviewer 2 Report

I consider the article to be of high quality. It can be concluded that the author (s) has done a large amount of work. The article is remarkable both from the point of view of theory and topicality of the topic. Methodologically, the work is done well. I particularly appreciate the choice of a topic that is focused on an issue that is relevant today within the framework of religious-political discussions. In the context of current trends, this article brings insights into the constant need to discover problems that are connected to value searches and orientation in today's world. The work is well structured, the abstract is concise and long enough. The conclusions correspond to the corpus of the text. The reference literature is adequate and its number as well as the time horizon are in order.

The author made the adjustments I recommended to the text. From the material point of view, the text is suitable for publication.

Only technical details remain in the text, which need to be edited.

It is necessary to unify the font size in the footnotes.

Looking at the text, it looks like there are unnecessary spaces in the sentences (It's a pdf text, I could be wrong).

These are the lines:

96:   ...  consciousness.  X    Secular nationalism ...

198: ... endangered identity   X  (Posern-ZieliÅ„ski 2005, p. 25).

212: ... religion as such.  X  Political ...

282: ... group identity  X  (…) which with ...

331: ... after WWII,  X  it was not  ...

420: ... 2022, p. 8),  X   this may ...

494: ... p. 26)  X   turned out ...

533: ... modernization  X   (Casanova 2004 ...

536: ... exceptionally.  X   Strategic, political ...

After editing the technical details, I recommend the article for publication.

Author Response

Dear Reviewer, once again thank you for your valuable comments and suggestions. I have to admit I am impressed with how careful and detailed your revision of my article was including the editiorial dimension.